# Neuromuscular blockade in mechanically ventilated pneumonia patients with moderate to severe hypoxemia: A multicenter retrospective study

**Moon Seong Baek[1]◉, Jong Ho Kim[2,3]◉, Yaeji Lim[4], Young Suk Kwon [2,3]***

**1** Department of Internal Medicine, Chung-Ang University Hospital, Chung-Ang University College of Medicine, Seoul, Republic of Korea, **2** Department of Anesthesiology and Pain Medicine, College of Medicine, Hallym University, Chuncheon Sacred Heart Hospital, Chuncheon, Republic of Korea, **3** Institute of New Frontier Research Team, Hallym University, Chuncheon, South Korea, **4** Department of Applied Statistics, Chung-Ang University, Seoul, Republic of Korea

◉ These authors contributed equally to this work.

* gettys@hallym.or.kr

**Data Availability Statement:** https://doi.org/10.6084/m9.figshare.21391353.v1.

**Funding:** The author(s) received no specific funding for this work.

## Abstract

### Background/Purpose

The benefit of neuromuscular blockades (NMBs) in critically ill patients receiving mechanical ventilation remains uncertain. Therefore, we aimed to investigate whether NMB use is associated with improved survival of mechanically ventilated pneumonia patients with moderate to severe hypoxemia.

### Methods

This retrospective multicenter study was conducted at five university-affiliated hospitals. Data of pneumonia patients aged 18 years and older who received mechanical ventilation between January 1, 2011, and December 31, 2020, were analyzed.

### Results

In a total of 1,130 patients, the mean patient age was 73.1 years (SD±12.6), and the overall mortality rate at 30 d was 29.5% (n = 333). NMB users had a higher 30 d mortality rate than NMB nonusers (33.9% vs. 26.8%, P = 0.014). After PS matching, the 30 d mortality rate was not significantly different between NMB users and nonusers (33.4% vs. 27.8%, p = 0.089). However, 90 d mortality rate was significantly increased in NMB users (39.7% vs. 31.9%, p = 0.021). Univariable Cox proportional hazard regression analyses showed that NMB use ≥ 3 d was significant risk factor for the 90 d mortality than those with < 3 d use (90 d mortality HR 1.39 [95% CI: 1.01–1.91], P = 0.045).

### Conclusions

NMB use was not associated with lower 30 d mortality among mechanically ventilated pneumonia patients with moderate to severe hypoxemia. Rather, NMB users had higher 90 d

**Competing interests:** The authors have declared that no competing interests exist.

mortality, furthermore, and NMB use ≥ 3 d was associated with a higher risk of long-term mortality compared to NMB use < 3 d. Therefore, care should be taken to avoid extended use of NMB in critically ill pneumonia patients during mechanical ventilation.

## Introduction

During mechanical ventilation, neuromuscular blockades (NMBs) have been used in various critical care settings such as acute respiratory distress syndrome (ARDS) [1–5], severe sepsis [6,7], bronchospasm with profound hypoxemia [8], and therapeutic hypothermia [9,10]. The expected advantages of NMBs are a reduction of oxygen consumption, improvement of oxygenation, and optimization of mechanical ventilation through reducing patient–ventilator asynchrony [11]. However, it has been reported that the use of NMBs does not affect oxygen consumption in adequately sedated patients [12,13]. In addition, the use of NMBs may be associated with increased atelectasis, prolonged paralysis following NMB, and intensive care unit (ICU)-acquired weakness [11,14]. Therefore, NMBs should be used with caution in critically ill patients receiving mechanical ventilation.

It is well established that by reducing the risk of ventilator-induced lung injury, lung-protective mechanical ventilation can improve survival among patients with ARDS [15]. To facilitate lung-protective mechanical ventilation, a short course of NMB has been recommended in the early phase of moderate-to-severe ARDS [5,16,17]. In the ARDS et Curarisation Systematique (ACURASYS) trial, the early administration of a 48 h infusion of NMB in ARDS improved the adjusted 90 d survival for patients with a PaO$_2$/FiO$_2$ ratio < 150. Furthermore, a systematic review and meta-analysis using three randomized trials revealed that a short-term course of NMB was associated with reduced mortality and did not increase ICU-acquired weakness for patients with ARDS [4]. However, a recently reported trial, the Reevaluation of Systemic Early Neuromuscular Blockade (ROSE) trial, showed that there was no significant difference in 90 d mortality between patients who received an early cisatracurium infusion and those who were treated with a lighter sedation strategy [18]. With this finding contradicting the supposed efficacy of NMBs in ARDS, the benefit of NMBs remains uncertain in critically ill patients receiving mechanical ventilation. Therefore, we aimed to investigate whether NMB use is associated with improved survival among mechanically ventilated pneumonia patients with moderate to severe hypoxemia.

## Materials and methods

### Study design and patients

This retrospective multicenter study was conducted at five university-affiliated hospitals of Hallym University Medical Center in the Republic of Korea. The overall bed capacity and annual inpatients were approximately 3,000 beds (231 ICU beds) and 100,000 patients. We collected the electronic medical records of patients aged 18 years and older who received mechanical ventilation in the ICU between January 1, 2011, and December 31, 2020. During the study period, severe COVID-19 patients did not hospitalize at our ICUs. Among the mechanically ventilated patients identified, we enrolled patients with pneumonia. According to the clinical practice guideline of ARDS [19], NMBs have been used in moderate to severe hypoxemia to facilitate mechanical ventilation. Patients were excluded if they received surgery, underwent extracorporeal membrane oxygenation (ECMO) therapy, or had a duration of mechanical

ventilation < 3 d. We further excluded patients with $PaO_2$ (partial pressure of oxygen)/$FiO_2$ (fraction of inspired oxygen) $\geq$ 150 mmHg based on the initial arterial blood gas analysis. Patients received a bolus injection of NMB were excluded.

This study was approved by the Institutional Review Board of Chuncheon Sacred Heart Hospital (2021-03-011). The requirement for informed consent was waived owing to the retrospective nature of the analysis.

## Data collection and definitions

We obtained the following data on the day of initiation of mechanical ventilation through clinical big data analytic solution Smart CDW, which can analyze the electronic medical record text and integrated fixed data at five hospitals: patient age, sex, body mass index, diagnosis, time from hospitalization to ICU admission, time from hospitalization to mechanical ventilation initiation, Acute Physiology and Chronic Health Evaluation (APACHE) II score, modified early warning score, CURB-65, transfer from a skilled nursing facility, Charlson Comorbidity Index (CCI) and its variables, continuous renal replacement therapy, transfusion, vasopressors or inotropes, corticosteroids, opioids (morphine, fentanyl, or remifentanil), sedatives (propofol, midazolam, or dexmedetomidine), type and infused duration of NMBs (cisatracurium and vecuronium), and laboratory results with arterial blood gases.

In the assigned diagnoses for each patient, we retrieved pneumonia using the KCD-7 codes (Korean version of the International Classification of Diseases-10, ICD-10). Comorbidities were categorized by CCI variables according to the ICD-10 codes. The CURB-65 score is comprised of five separate elements, including confusion, urea (> 19 mg/dL), respiratory rate ($\geq$ 30/min), blood pressure (systolic blood pressure < 90 mmHg or diastolic blood pressure $\leq$ 60 mmHg diastolic), and age ($\geq$ 65 years) [20]. ARDS was identified as ICD-10 code of J80. To avoid inclusion of NMB users with bolus injection for endotracheal intubation purpose, we defined NMB users as patients who received continuous infusion of NMBs. The primary outcome was 30 d mortality from the initiation of mechanical ventilation. The secondary outcomes were 90 d mortality, length of stay, ICU stay, duration of mechanical ventilation, and the use of quetiapine and amiodarone.

## Statistical analysis

To construct a control group (NMB nonusers), propensity score (PS) matching was performed by the nearest-neighbor method with the following variables: age, sex, body mass index, time from hospitalization to ICU admission, time from hospitalization to mechanical ventilation initiation, APACHE II score, modified early warning score, CURB-65, transferred from skilled nursing facility, CCI, aspiration pneumonia, diabetes, congestive heart failure, myocardial infarction, chronic pulmonary disease, liver disease, moderate to severe chronic kidney disease, any malignancy, rheumatic disease, dementia, cerebrovascular disease, continuous renal replacement therapy, transfusion, vasopressors and inotropes, corticosteroids, opioids, sedatives, and $PaO_2$/$FiO_2$. In the matching process, 37 patients in the study groups were excluded due to null values of the matching variables.

Differences in continuous and categorical baseline characteristics between the groups were analyzed with Student's t-test and the chi-square test, respectively. Moreover, differences in baseline characteristics in the 1:1 matched group were analyzed with the paired t-test and McNemar's test. Univariable and multivariable stratified Cox proportional hazard regression models were used to calculate the hazard ratio (HR) and 95% confidence interval (CI) for the risk of mortality. Multivariate regression analysis was used to adjust the effect of imbalance in the baseline. Age, modified early warning score, Charlson comorbidity index, and positive end

expiratory pressure (PEEP) were selected as the adjust variables, because there were significant differences between two groups even after matching. Statistical analyses were performed using R 4.0.4 (The R Foundation for Statistical Computing).

## Results

### Patient characteristics

During the study period, 1,146,506 patients admitted to our hospitals, and 112,322 patients admitted to the ICU. Mechanically ventilated patients were 29,554, and patients with pneumonia aged over 18 years were 3,357 (Fig 1). We excluded 2,227 patients who received surgery (n = 404), had a mechanical ventilation duration < 3 d (n = 502), received ECMO (n = 58), had $PaO_2/FiO_2 \geq 150$ mmHg (n = 1,194), and received NMB as a bolus injection (n = 69). In a total of 1,130 patients, the mean patient age was 73.1 years (SD±12.6), and 67.2% were male. NMB users were 422 (37.3%), and the mean duration of NMB use was 4.6 d (SD±5.8). A comparison of the patients' baseline characteristics is presented in Table 1. Age, CCI, and $PaO_2/FiO_2$ ratio were higher in NMB nonusers than NMB users. On the other hand, modified early warning score, PEEP and the use corticosteroids, opioids, and sedatives were higher in NMB users. There were no significant differences in the APACHE II score and continuous renal replacement therapy.

### Outcomes

Overall mortalities at 30 d and 90 d were 29.5% (n = 333) and 35.1% (n = 397), respectively.

NMB users had significantly higher 30 d and 90 d mortalities than NMB nonusers (30 d mortality 33.9% vs. 26.8%, P = 0.014; 90 d mortality 40.0% vs. 32.2%, P = 0.009) (Table 2). ICU stay and duration of mechanical ventilation were significantly longer in the NMB users than the NMB nonusers. The incidence of tracheostomy was significantly higher in the NMB users (tracheostomy 28.7% vs. 11.9%, p<0.001). After PS matching, 30 d mortality was not significantly different between NMB users and nonusers (33.4% vs. 27.8%, p = 0.089). However, 90 d mortality was significantly higher in the NMB users (39.7% vs. 31.9%, p = 0.021).

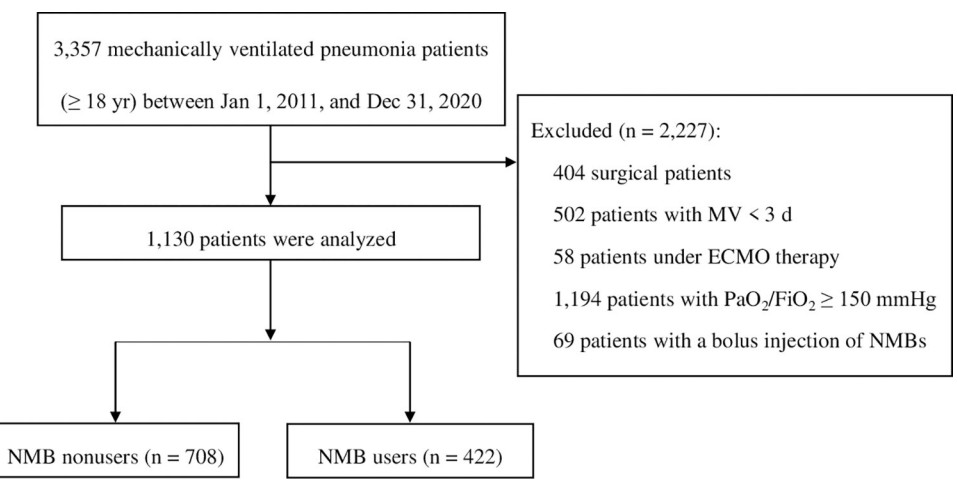

**Fig 1. Patient flowchart.** NMB, neuromuscular blockade; MV, mechanical ventilation; ECMO, extracorporeal membrane oxygenation; PaO$_2$, partial pressure of oxygen; and FiO$_2$, fraction of inspired oxygen.

**Table 1. Comparison of baseline characteristics of patients.**

| Variables | Before matching | | | After matching | | | |
| --- | --- | --- | --- | --- | --- | --- | --- |
| | NMB nonuser (n = 708) | NMB user (n = 422) | P value | NMB nonuser (n = 395) | NMB user (n = 395) | P value | Standardized mean difference |
| Age (years) | 74.3±12.4 | 71.1±12.8 | <0.001 | 72.7±12.9 | 71.0±12.9 | 0.058 | 0.130 |
| Male sex (%) | 465 (65.7) | 294 (69.7) | 0.188 | 274 (69.4) | 277 (70.1) | 0.875 | 0.017 |
| Body mass index (kg/m$^2$) | 21.5±4.4 | 22.0±4.3 | 0.054 | 21.9±4.7 | 22.0±4.3 | 0.665 | 0.030 |
| Time from hospitalization to ICU admission (d) | 1.8±5.8 | 1.8±4.6 | 0.986 | 1.5±4.3 | 1.6±3.9 | 0.592 | 0.037 |
| Time from hospitalization to MV initiation (d) | 4.5±16.5 | 3.8±6.5 | 0.316 | 3.5±7.2 | 3.5±5.8 | 0.974 | 0.002 |
| APACHE II score | 24.9±5.7 | 25.4±5.9 | 0.173 | 25.3±5.8 | 25.4±5.9 | 0.716 | 0.025 |
| Modified early warning score | 5.5±1.7 | 5.9±1.8 | <0.001 | 5.7±1.7 | 5.9±1.8 | 0.039 | 0.144 |
| CURB-65 | 2.8±0.9 | 2.7±1.0 | 0.147 | 2.8±0.9 | 2.7±1.0 | 0.509 | 0.047 |
| Transferred from skilled nursing facility (%) | 192 (27.1) | 68 (16.1) | <0.001 | 79 (20.0) | 63 (15.9) | 0.160 | 0.106 |
| Charlson Comorbidity Index | 4.8±2.3 | 4.4±2.2 | 0.001 | 4.6±2.2 | 4.3±2.2 | 0.081 | 0.117 |
| Aspiration pneumonia (%) | 145 (20.5) | 53 (12.6) | 0.001 | 57 (14.4) | 47 (11.9) | 0.337 | 0.075 |
| Comorbidities[a] (%) | | | | | | | |
| Diabetes | 150 (21.2) | 83 (19.7) | 0.593 | 81 (20.5) | 73 (18.5) | 0.837 | 0.051 |
| Congestive heart failure | 113 (16.0) | 56 (13.3) | 0.254 | 61 (15.4) | 52 (13.2) | 0.402 | 0.065 |
| Myocardial infarction | 34 (4.8) | 22 (5.2) | 0.868 | 23 (5.8) | 21 (5.3) | 0.877 | 0.022 |
| Chronic pulmonary disease | 151 (21.3) | 94 (22.3) | 0.765 | 87 (22.0) | 88 (22.3) | 1.000 | 0.006 |
| Liver disease | 48 (6.8) | 35 (8.3) | 0.409 | 34 (8.6) | 35 (8.9) | 1.000 | 0.009 |
| Moderate to severe CKD | 88 (12.4) | 35 (8.3) | 0.039 | 36 (9.1) | 31 (7.8) | 0.603 | 0.045 |
| Any malignancy | 66 (9.3) | 52 (12.3) | 0.135 | 43 (10.9) | 47 (11.9) | 0.731 | 0.032 |
| Rheumatic disease | 9 (1.3) | 19 (4.5) | 0.001 | 9 (2.3) | 17 (4.3) | 0.153 | 0.114 |
| Dementia | 92 (13.0) | 39 (9.2) | 0.070 | 40 (10.1) | 35 (8.9) | 0.620 | 0.043 |
| Cerebrovascular disease | 194 (27.4) | 77 (18.2) | 0.001 | 97 (24.6) | 70 (17.7) | 0.018 | 0.168 |
| Continuous renal replacement therapy (%) | 59 (8.3) | 36 (8.5) | 0.996 | 33 (8.4) | 36 (9.1) | 0.798 | 0.027 |
| Transfusion (%) | 110 (15.5) | 80 (19.0) | 0.160 | 60 (15.2) | 72 (18.2) | 0.271 | 0.082 |
| Vasopressors and inotropes (%) | 304 (42.9) | 206 (48.8) | 0.063 | 190 (48.1) | 197 (49.9) | 0.664 | 0.035 |
| Corticosteroids (%) | 82 (11.6) | 84 (19.9) | <0.001 | 62 (15.7) | 78 (19.7) | 0.118 | 0.106 |
| Opioids (%) | 516 (72.9) | 357 (84.6) | <0.001 | 320 (81.0) | 335 (84.8) | 0.164 | 0.101 |
| Sedatives (%) | 443 (62.6) | 380 (90.0) | <0.001 | 348 (88.1) | 355 (89.9) | 0.337 | 0.057 |
| PaO$_2$/FiO$_2$ (mmHg) | 91.6±30.7 | 87.6±30.8 | 0.036 | 90.9±30.9 | 87.5±30.8 | 0.125 | 0.110 |
| PEEP | 7.1±2.6 | 8.2±3.1 | <0.001 | 7.5±2.8 | 8.2±3.1 | 0.001 | 0.239 |

Values are presented as mean±SD or n (%).

[a] The CURB-65 score is comprised of five separate elements including confusion, uremia, respiratory rate, blood pressure, and age ≥ 65 years.

[b] Comorbidities were categorized by the Charlson Comorbidity Index.

NMB, neuromuscular blockade; ICU, intensive care unit; MV, mechanical ventilation; APACHE, Acute Physiology and Chronic Health Evaluation; CKD, chronic kidney disease; PaO$_2$, arterial partial pressure of oxygen; FiO$_2$, fraction of inspired oxygen; and PEEP, positive end expiratory pressure.

In the Kaplan–Meier survival analyses, there was no significant difference in 30 d mortality (p = 0.152), however, 90 d mortality was significantly higher in the NMB users (p = 0.047) (Fig 2). In the multivariate Cox regression analysis, NMB use was not independently associated with a higher risk of mortality: HRs for 30 d and 90 d mortalities according to NMB use were 1.06 ([95% CI: 0.82–1.37], P = 0.659) and 1.14 ([95% CI: 0.90–1.44], P = 0.294), respectively (Table 3).

**Table 2. Clinical outcomes.**

|  | Before matching | | | After matching | | |
| --- | --- | --- | --- | --- | --- | --- |
| Variables | NMB nonuser (n = 708) | NMB user (n = 422) | P value | NMB nonuser (n = 395) | NMB user (n = 395) | P value |
| Mortality at 30 d (%) | 190 (26.8) | 143 (33.9) | 0.014 | 110 (27.8) | 132 (33.4) | 0.089 |
| Mortality at 90 d (%) | 228 (32.2) | 169 (40.0) | 0.009 | 126 (31.9) | 157 (39.7) | 0.021 |
| Length of stay (d) | 31.1±34.9 | 32.5±28.0 | 0.440 | 31.7±34.9 | 32.8±28.5 | 0.621 |
| Length of ICU stay (d) | 19.3±20.8 | 23.4±22.5 | 0.002 | 19.9±23.0 | 23.6±23.0 | 0.020 |
| Duration of MV (d) | 13.4±15.7 | 18.6±19.6 | <0.001 | 14.0±17.4 | 18.8±20.1 | <0.001 |
| Tracheostomy (%) | 84 (11.9) | 121 (28.7) | <0.001 | 52 (13.2) | 112 (28.4) | <0.001 |
| Chest tube insertion (%) | 43 (6.1) | 28 (6.6) | 0.803 | 29 (7.3) | 26 (6.6) | 0.775 |
| Use of quetiapine (%) | 150 (21.2) | 108 (25.6) | 0.102 | 91 (23.0) | 103 (26.1) | 0.346 |
| Use of amiodarone (%) | 81 (11.4) | 63 (14.9) | 0.108 | 48 (12.2) | 60 (15.2) | 0.271 |

Values are presented as mean±SD or n (%).

NMB, neuromuscular blockade; ICU, intensive care unit; MV, mechanical ventilation.

## Subgroup analysis

In NMB users, there was no significant difference in 30 d mortality (p = 0.055), however, 90 d mortality was significantly higher in the NMB users (p = 0.045) (Fig 3). Multivariable Cox proportional hazard regression analyses showed that NMB use ≥ 3 d was not a predictor of mortality: HRs for 30 d and 90 d mortalities according to NMB use were 1.30 ([95% CI: 0.91–1.85], P = 0.147) and 1.31 ([95% CI: 0.95–1.80], P = 0.105), respectively (Table 3).

## Discussion

This multicenter study revealed that NMB use was not associated with lower 30 d mortality in mechanically ventilated pneumonia patients with moderate to severe hypoxemia. However, 90 d mortality was significantly increased in NMB users. NMB users had a longer duration of mechanical ventilation and/or ICU stay, and underwent tracheostomy more frequently than NMB nonusers. Furthermore, NMB use ≥ 3 d was associated with higher 90-d mortality rate compared to NMB use < 3 d.

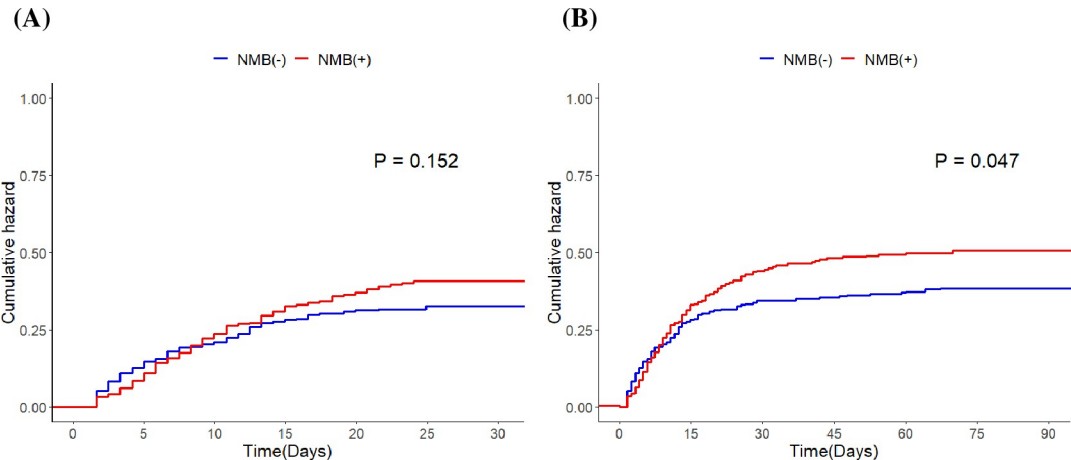

**Fig 2.** Kaplan–Meier survival analyses (A) 30 d mortality according to NMB use. (B) 90 d mortality according to NMB use. NMB, neuromuscular blockade.

**Table 3. Univariate and multivariate Cox regression analyses for 30 d and 90 d mortalities.**

| | | | Univariate model | | Multivariate model | |
|---|---|---|---|---|---|---|
| 30 d mortality | No. of patients | No. of events | HR (95% CI) | P value | HR (95% CI) | P value |
| NMB nonusers | 395 | 110 | 1 (reference) | | 1 (reference) | |
| NMB users | 395 | 132 | 1.20 (0.93–1.55) | 0.152 | 1.06 (0.82–1.37) | 0.659 |
| NMB < 3 d | 183 | 51 | 1 (reference) | | 1 (reference) | |
| NMB ≥ 3 d | 212 | 81 | 1.41 (0.99–2.00) | 0.055 | 1.30 (0.91–1.85) | 0.147 |
| 90 d mortality | No. of patients | No. of events | HR (95% CI) | P value | HR (95% CI) | P value |
| NMB nonusers | 395 | 126 | 1 (reference) | | 1 (reference) | |
| NMB users | 395 | 157 | 1.27 (1.00–1.60) | 0.047 | 1.14 (0.90–1.44) | 0.294 |
| NMB < 3 d | 183 | 62 | 1 (reference) | | 1 (reference) | |
| NMB ≥ 3 d | 212 | 95 | 1.39 (1.01–1.91) | 0.045 | 1.31 (0.95–1.8) | 0.105 |

The multivariable analysis included age, sex, body mass index, time from hospitalization to ICU admission, time from hospitalization to MV initiation, APACHE II score, modified early warning score, CURB-65, transferred from skilled nursing facility, Charlson Comorbidity Index, aspiration pneumonia, diabetes, congestive heart failure, myocardial infarction, chronic pulmonary disease, liver disease, moderate to severe CKD, any malignancy, rheumatic disease, dementia, cerebrovascular disease, continuous renal replacement therapy, transfusion, vasopressors and inotropes, corticosteroids, opioids, sedatives, and PaO$_2$/FiO$_2$ as the covariates.

NMB, neuromuscular blockade; HR, hazard ratio; CI, confidence interval.

Based on the results of the ROSE trial and clinical studies on sedation [18,21–23], a recent practice guideline does not recommend the routine use of NMBs, even in moderate or severe ARDS [24]. This change of strategy in patients with ARDS can cause confusion for physicians regarding the use of NMBs in patients with pneumonia, which is the major cause of acute respiratory failure in the ICU [25]. In this regard, the results of our cohort provide valuable information that the use of NMBs during mechanical ventilation did not have a survival benefit in patients with severe hypoxemic pneumonia. NMBs tended to be administered to more critically ill patients because NMBs are often used in a setting where a lighter sedation strategy does not make the mechanical ventilation tolerable. We matched the clinical variables that can influence the outcomes between NMB users and nonusers as much as possible. Even after PS matching, 30 d mortality rate was not significantly different between NMB users and nonusers.

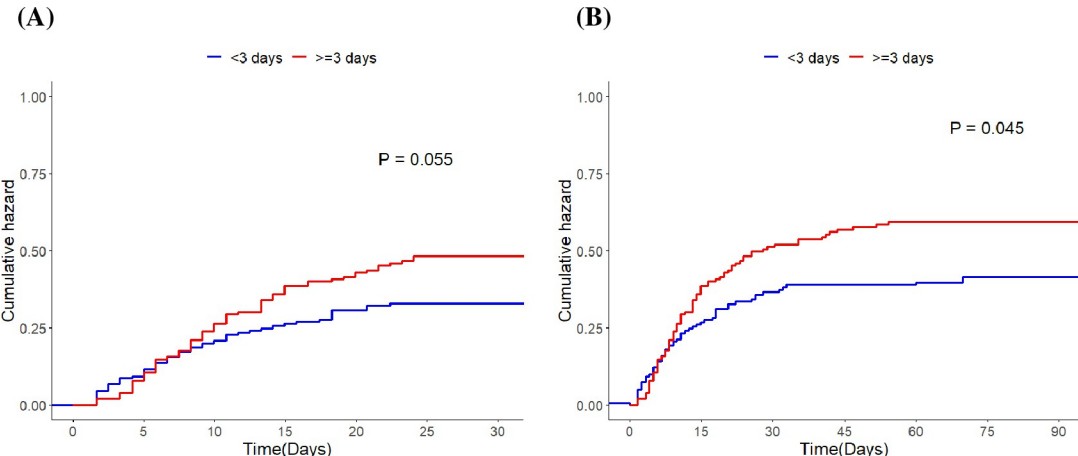

**Fig 3.** Kaplan–Meier survival analyses (A) 30 d mortality according to the duration of NMB use. (B) 90 d mortality according to the duration of NMB use. NMB, neuromuscular blockade.

Rather, 90 d mortality rate was significantly higher in NMB users. Therefore, NMBs during mechanical ventilation should be used cautiously in hypoxemic pneumonia patients.

In a recently reported meta-analysis with five randomized controlled trials, NMBs did not reduce the risk of death at 28 days (RR, 0.90; 95% CI, 0.78–1.03, P = 0.12) in ARDS patients [26]. Deep sedation accompanied by NMBs can result in negative outcomes such as mortality or delayed time to extubation [27]. It has been reported that early deep sedation is associated with higher mortality in mechanically ventilated critically ill patients [21–23,27]. Shehabi et al. suggested that the sedation intensity in the first 48 h was an independent predictor of increased 180 d mortality (HR [95% CI], 1.29 [1.15–1.46], p<0.001) [21]. In the multivariable Cox proportional hazard regression models, depth of sedation was positively correlated with mortality. They suggested that light sedation could be a goal to improve patients' outcomes in the early stage of mechanical ventilation [21,22,27]. Although we did not present the data of sedation intensity, there is a possibility that NMB users were more deeply sedated than NMB nonusers. Therefore, NMBs should be avoided whenever possible in critically ill patients who can tolerate ventilation with a lighter sedation according to recent ARDS guidelines [24].

NMBs have some favorable effects in patients with ARDS. NMB infusion is associated with an improvement in oxygenation and can be beneficial in increasing expiratory transpulmonary pressure in moderate to severe ARDS patients [28]. Ho et al. reported that the $PaO_2/FIO_2$ ratio at 48 h in moderate to severe ARDS patients was higher in NMB users than nonusers, and the risk of barotrauma was lower in NMB users than nonusers (RR 0.55 [95% CI: 0.35–0.85], P = 0.007) [26]. On the other hand, NMB use is associated with poor outcomes such as delayed extubation, increased delirium, and a longer duration of mechanical ventilation [21–23,27,29]. In accordance with previous studies, our results revealed that NMB users had longer durations of ICU stay and mechanical ventilation. These may be associated with the higher incidence of tracheostomy in NMB users.

Concerns regarding NMB use include not only early deep sedation but also the prolonged use of NMBs. According to recent guidelines and clinical studies, the use of NMBs for up to 48 h is recommended [3,18,24,30,31]. Dodson et al. reported that long-term NMB infusion is associated with denervation-like changes, which could cause prolonged muscle paralysis [32]. Saccheri et al. reported that ICU-acquired weakness was associated with increased long-term mortality in critically ill patients [33]. It has been suggested that NMB use ≥ 3 d can cause ICU-acquired weakness, and this may lead to increased 90 d mortality rates. Although the association between NMB use and ICU-acquired weakness remains controversial [34], we found that NMB users had a 1.4 times higher mortality rate at 90 days than NMB nonusers. If NMB is required to achieve lung protective ventilation, it is desirable to discontinue it within 3 days if possible.

This study has some limitations. First, due to the retrospective nature of the study, it was difficult to obtain some information. We did not describe the exact doses of drugs or adjuvant therapy such as prone positioning, which might have affected mortality. In addition, ventilatory parameters and respiratory mechanics such as driving pressure, plateau pressure, tidal volume, and static compliance were not collected. NMB can be used with various indications in critically ill patients such as facilitation of tracheal intubation, facilitation of mechanical ventilation, control of status asthmaticus or therapeutic hypothermia [14]. Although, we cannot demonstrate exact indication of NMB, patients who used NMB for intubation purpose would not be included in NMB users. Second, although PS methods were employed to match the potential differences between the two groups, there might have been residual confounding factors. Therefore, additional clinical studies are needed to assess the effect of NMBs in critically ill patients receiving mechanical ventilation for reasons other than ARDS.

## Conclusions

NMB use was not associated with lower 30 d mortality in mechanically ventilated pneumonia patients with moderate to severe hypoxemia. Rather, NMB users had higher 90 d mortality and longer duration of mechanical ventilation or ICU stay than NMB nonusers. Furthermore, NMB use ≥ 3 d was associated with higher 90 d mortality compared to NMB use < 3 d. Therefore, NMBs should be used with caution in critically ill pneumonia patients, and prolonged NMB use should be avoided if possible.

## Author Contributions

**Data curation:** Jong Ho Kim, Yaeji Lim.

**Formal analysis:** Moon Seong Baek, Yaeji Lim.

**Investigation:** Jong Ho Kim, Yaeji Lim.

**Methodology:** Moon Seong Baek, Young Suk Kwon.

**Software:** Jong Ho Kim.

**Supervision:** Young Suk Kwon.

**Visualization:** Jong Ho Kim.

**Writing – original draft:** Moon Seong Baek, Young Suk Kwon.

**Writing – review & editing:** Moon Seong Baek, Yaeji Lim, Young Suk Kwon.

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
