## [Decision Letter · Decision Letter 0]

22 Aug 2022

PONE-D-21-37840Neuromuscular blockade in mechanically ventilated pneumonia patients with moderate to severe hypoxemiaPLOS ONE

Dear Dr. Kwon,

Thank you for submitting your manuscript to PLOS ONE. After careful consideration, we feel that it has merit but does not fully meet PLOS ONE’s publication criteria as it currently stands. Therefore, we invite you to submit a revised version of the manuscript that addresses the points raised during the review process. Academic Editor - Thank you for submitting your paper to us for review.  I sent it to five distinguished referees for comment and decision of whom two agreed to review; you will see these below.  They thought that the paper has merit, but each have raised some substantial issues to be addressed in a revision.  Please carefully consider the comments below and reply directly to each in a cover letter with appropriate marked and linked changes to the manuscript.  I look forward to seeing the revision, which I will send back to the same referees for further comment and decision.  Please understand that this is not a guarantee of future publication, as the revised manuscript itself must stand on its own merit.  In particular, please pay attention to the comments from the first referee in regard to COVID.

We look forward to receiving your revised manuscript.

Kind regards,

Steven Eric Wolf, MD

Academic Editor

PLOS ONE

Journal Requirements:

Reviewers' comments:

Reviewer's Responses to Questions

**Comments to the Author**

1. Is the manuscript technically sound, and do the data support the conclusions?

Reviewer #1: Yes

Reviewer #2: Yes

2. Has the statistical analysis been performed appropriately and rigorously? 

Reviewer #1: Yes

Reviewer #2: I Don't Know

3. Have the authors made all data underlying the findings in their manuscript fully available?

Reviewer #1: Yes

Reviewer #2: No

4. Is the manuscript presented in an intelligible fashion and written in standard English?

Reviewer #1: Yes

Reviewer #2: No

5. Review Comments to the Author

Reviewer #1: 1)The time frame of this study includes the COVID pandemic. There is no reference to these patients in the paper. Were they excluded? They qualify as pneumonia. COVID patients do not start out with P/F ratios <150, but may progress there as they go into ARDS

2) ADRS vs severe pneumonia is not differentiated. Was this on purpose? How does your diagnosis of severe pneumonia differ from ARDS criteria?

3) Capacity of the institutions state the overall beds in the hospitals. For this paper, it would be useful to also identify how may ICU beds were available.

4) When excluding surgical patients, how was this determined? Any and all surgical procedures (including tracheostomy before inclusion) or just major surgical intervention. There is a good population of surgical patients who suffer from pneumonia post op.

5) There are many different indications to use neuromuscular blockade listed in the manuscript, but the triggers to paralyze are not clear. It would help if the criteria could be defined

Reviewer #2: Thank you for the opportunity to review this interesting manuscript on a relevant theme. The authors conducted a retrospective cohort study to evaluate whether neuromuscular blockade use was associated with improved survival among mechanically ventilated pneumonia patients with moderate to severe hypoxemia. The study's main limitation was that ventilatory parameters and respiratory mechanics were not evaluated between neuromuscular blockade users and nonusers, including driving pressure, plateau pressure, PEEP, tidal volume, and static complacence.

There are some questions and comments below that I hope may contribute to strengthening the paper.

#1 - The authors should indicate that the study is a retrospective cohort.

#2 - The authors should provide full detail of the population sampling technique from each hospital and include both unweight sampling number, weight %, and standard errors.

#3 - For the Cox proportional hazard regression analysis, the authors should clarify the methods for variable selection in the model.

#4 - Were there missing data? If yes, the authors should include complete details of how they handled missing data.

#5 - Regarding propensity, matching variables should be selected based on their theoretical association with both the outcome and treatment group assignment. Further information should be included in the methods to replicate and justify the matched sample. Which model (logit/probit regression) was used for the propensity score? Additionally, please provide propensity score analysis estimates and their precision, e.g., 95% confidence interval. Was there any iteration in the variable selection for the propensity scores?

#6 - Please clarify the working definition used for acute respiratory distress syndrome in the Methods. Was it Berlin definition of acute respiratory distress syndrome? Were patients with COVID-19 diagnosis included in the study?

6. PLOS authors have the option to publish the peer review history of their article (what does this mean?). If published, this will include your full peer review and any attached files.

Reviewer #1: No

Reviewer #2: No

---

## [Author Response · Author response to Decision Letter 0]

20 Sep 2022

September 20, 2022

Manuscript ID Number: PONE-D-21-37840

Neuromuscular blockade in mechanically ventilated pneumonia patients with moderate to severe hypoxemia: a multicenter retrospective study

Dear Editor

Thank you for your careful reviewing our manuscript and giving valuable comments. We have made some corrections and modification following the helpful comments. I am enclosing a list of the modification of the manuscript that we have made and our replies to the comments. 

We believe we have addressed all questions and comments, but please give us more opportunity to respond if there is anything unsatisfactory or questionable.

We thank you and reviewers again for the constructive review. We are looking forward to hearing positive reply from you soon.

Sincerely yours,

Young Suk Kwon, MD

Department of Anesthesiology and Pain Medicine, Chuncheon Sacred Heart Hospital, 77 Sakju-ro, Chuncheon, 24253, South Korea

Tel: +82-10-8745-7514

Response to Reviewer 1

Q1) The time frame of this study includes the COVID pandemic. There is no reference to these patients in the paper. Were they excluded? They qualify as pneumonia. COVID patients do not start out with P/F ratios <150, but may progress there as they go into ARDS

A1) Thank you for your thoughtful comments. In this study, we enrolled the patients until December 31, 2020. Per your concern, it is a time that overlaps with the COVID pandemic. However, Korean government has prepared and responded effectively to COVID-19 crisis in the early stage. From November 2021, deaths due to severe COVID-19 was increased, executive order by Korean government was performed for preparedness of ICU beds for patients with severe COVID-19. Therefore, our five hospitals started isolation beds for critically ill COVID-19 patients at this time. So, we did not include COVID-19 patients in our study population. We have revised the manuscript as follows:

Materials and methods (revised manuscript, page 4 lines 11)

During the study period, severe COVID-19 patients were not hospitalized in the ICU.

Q2) ADRS vs severe pneumonia is not differentiated. Was this on purpose? How does your diagnosis of severe pneumonia differ from ARDS criteria?

A2) We aimed to investigated the effect of NMB in mechanically ventilated “pneumonia patients”. And pneumonia is one of the common cause of ARDS, so these could be overlapped. We did not differentiate the diseases. However, per your suggestion, we identified 27 patients with ARDS diagnosis via ICD-10 codes of diagnosis (J80). Therefore, we excluded these patients, and statistical analyses were conducted again with the help of Statistics Department of Chung-Ang University. Therefore, All Tables and Figures were revised. We have revised the flow-chart as follows: 

Figure 1. Flowchart (revised manuscript, page 7 lines 13~15)

Materials and methods (revised manuscript, page 4 lines 18~19)

Patients who received a bolus injection of NMBs and with a diagnosis of ARDS were excluded.

Results (revised manuscript, page 7 lines 2~3)

had a diagnosis of ARDS (n = 27), and received a bolus injection of NMBs (n = 69).

Q3) Capacity of the institutions state the overall beds in the hospitals. For this paper, it would be useful to also identify how may ICU beds were available.

A3) Our hospitals included 231 ICU beds. Per your suggestion, we have revised the manuscript as follows:

Materials and methods (revised manuscript, page 4 lines 7~9)

The overall bed capacity and annual number of inpatients were approximately 3,000 beds (231 ICU beds) and 100,000 patients, respectively.

Q4) When excluding surgical patients, how was this determined? Any and all surgical procedures (including tracheostomy before inclusion) or just major surgical intervention. There is a good population of surgical patients who suffer from pneumonia post op.

A4) Except for 69 tracheostomy before inclusion, all surgical patients were excluded. Per your comments, there is a possibility that surgical patients can be underwent pneumonia as post op complication. However, this retrospective study can not identify the exact cause of pneumonia in surgical patients. Moreover, surgical patients may have trauma, and this can influence to the mortality, hence surgical patients can be potentially heterogeneous. In addition, surgical patients can receive mechanical ventilation and neuromuscular blockades for the surgery. For these reasons, we excluded surgical patients.

Q5) There are many different indications to use neuromuscular blockade listed in the manuscript, but the triggers to paralyze are not clear. It would help if the criteria could be defined

A5) Per your comments, there are several indications of neuromuscular blockade. According to the Renew et al., neuromuscular blockade has been used for the facilitation of tracheal intubation and facilitation of mechanical ventilation in critically ill patients (Renew et al. Journal of Intensive Care (2020) 8:37). Status asthmaticus or therapeutic hypothermia can be indications for the use of neuromuscular blockades. However, we cannot demonstrate the clear indication fo neuromuscular blockade because of the nature of retrospective study. Generally, neuromuscular blockades used bolus injection during endotracheal intubation. Therefore, we defined neuromuscular blocker user as cisatracrium and vecuronium via continuous infusion, and 69 patients received NMB via injection was excluded. not injection mehod. Although, we cannot demonstrate exact indication of neuromuscular blockade, patients who used neuromuscular blockade for intubation purpose would not be included in NMB users. Per your concern, we further described this in the limitation section. We have revised the manuscript as follows:

Materials and methods (revised manuscript, page 4 lines 18~19)

Patients who received a bolus injection of NMBs and with a diagnosis of ARDS were excluded.

Materials and methods (revised manuscript, page 5 lines 17~18)

To avoid including NMB users requiring a bolus injection for endotracheal intubation, we defined NMB users as patients who received a continuous infusion of NMBs.

Results (revised manuscript, page 7 lines 2~3)

had a diagnosis of ARDS (n = 27), and received a bolus injection of NMBs (n = 69).

Discussion (revised manuscript, page 13 lines 9~13)

NMBs may be used for various purposes in critically ill patients, such as facilitation of tracheal intubation, facilitation of mechanical ventilation, control of status asthmaticus, or therapeutic hypothermia.[18] Although we could not identify the exact indications for NMB use, patients who used NMBs for intubation were not included in the group of NMB users in this study.

 

Response to Reviewer 2

Reviewer #2: Thank you for the opportunity to review this interesting manuscript on a relevant theme. The authors conducted a retrospective cohort study to evaluate whether neuromuscular blockade use was associated with improved survival among mechanically ventilated pneumonia patients with moderate to severe hypoxemia. The study's main limitation was that ventilatory parameters and respiratory mechanics were not evaluated between neuromuscular blockade users and nonusers, including driving pressure, plateau pressure, PEEP, tidal volume, and static complacence. 

A) Thank you for your thoughtful commens. Unfourtunately, several parameters such as driving pressure, plateau pressure or tidal volume did not obtained due to the retrospective study design. There are a lot of missing data in these variables. However, per your suggestion, we added PEEP varable in this revision, and we have revised the limitation section as follows:

Moreover, per Reviewr #1’s suggestion, we identified 27 patients with ARDS and 69 bolus injection of NMB, and excluded them. All statistical analysis was conducted again with the help of Statistics Department of Chung-Ang University. Therefore, Tables and Figures were revised.

Discussion (revised manuscript, page 13 lines 7~9)

In addition, ventilatory and respiratory parameters such as driving pressure, plateau pressure, tidal volume, and static compliance were not collected.

There are some questions and comments below that I hope may contribute to strengthening the paper.

Q1) The authors should indicate that the study is a retrospective cohort.

A1) Thank you for your thoughtful comment. Per your suggestion, we have revised the title as follows:

Title

Neuromuscular blockade in mechanically ventilated pneumonia patients with moderate to severe hypoxemia: a multicenter retrospective study

Q2) The authors should provide full detail of the population sampling technique from each hospital and include both unweight sampling number, weight %, and standard errors.

A2) Hallym University Medical Center (HUMC) had clinical big data analytic solution Smart CDW, which can analyze the electronic medical record text and integrated fixed data at five hospitals. Thus, using the Smart CDW, we collected clinical data of total patients with pneumonia at five hospitals. 

During the study period (between Jan. 1, 2011, and Dec. 31, 2020), 1,146,506 patients admitted to hospitals. Patients who admitted to the ICU were 112,322, and mechanically ventilated patients were 29,554. Among them, patients with pneumonia (≥18 yr) were 3,357. We have revised the manuscript as follows:

Materials and methods (revised manuscript, page 5 lines 1~3)

We used a clinical big data analytic solution (Smart CDW), which can analyze the electronic medical record text and integrate the data of five hospitals. The following data on the day of initiation of mechanical ventilation were obtained

Results (revised manuscript, page 6 lines 22~24)

During the study period, 1,146,506 patients were admitted to the hospitals, and 112,322 patients were admitted to the ICU. The number of mechanically ventilated patients was 29,554, and there were 3,357 patients with pneumonia aged over 18 years (Fig 1).

Q3) For the Cox proportional hazard regression analysis, the authors should clarify the methods for variable selection in the model.

A3) In this paper, univariable and multivariable stratified Cox proportional hazard regression models were used to compare the survival between two groups. Please note that we used stratified Cox proportional hazard regression, because it is a proper method for the matched (correlated) data set. Multivariate regression analysis was used to adjust the effect of imbalance in the baseline. Age, Modified early warning score, Charlson comorbidity index, and PEEP were selected as the adjust variables, because there were significant differences between two groups even after matching. We have revised the manuscript as follows:

Materials and methods (revised manuscript, page 6 lines 12~17)

Univariable and multivariable stratified Cox proportional hazard regression models were used to calculate the hazard ratio (HR) and 95% confidence interval (CI) for the risk of mortality. Multivariate regression analysis was used to adjust the effect of imbalance in the baseline. Age, modified early warning score, CCI, and positive end expiratory pressure (PEEP) were selected as adjusted variables as there were significant differences between two groups even after matching.

Q4) Were there missing data? If yes, the authors should include complete details of how they handled missing data.

A4) There were total 1103 patients in this data set, and 411 patients used NMB. The number of missing values in each variable is as follows.

Number of missing observations : PEEP - 5 , BMI - 39, APACHE - 45

Several studies recommend multiple imputation to impute missing data, but there is still debate as to which method is the most appropriate for propensity score matching [1,2]. Even we excluded the missing values in the matching process, only 27 (6.5%) patients of the NMB users are excluded from the original data set and 384 NMB users are obtained after matching. Therefore, we did not consider any imputation method in this study.

1. Stuart EA. Matching methods for causal inference: A review and a look forward. Stat Sci 2010;25:1-21.

2. Leite W, Aydin B. editors. A comparison of methods for imputation of missing covariate data prior to propensity score analysis. Washington, DC: American Education Research Association Conference, 2016.

Q5) Regarding propensity, matching variables should be selected based on their theoretical association with both the outcome and treatment group assignment. Further information should be included in the methods to replicate and justify the matched sample. Which model (logit/probit regression) was used for the propensity score? Additionally, please provide propensity score analysis estimates and their precision, e.g., 95% confidence interval. Was there any iteration in the variable selection for the propensity scores?

A5) Thank you for the comment. 

Propensity score analysis estimates and their precision (standard error) are presented in the following table (Table 1). As you pointed out, matching variables can be selected based on the theoretical association with both the outcome and treatment group assignment. However, some variables should be included as matching variables based on the previous clinical research even if they were not statistically significant. Therefore, in this paper we selected matching variables based on both theoretical association in our data and also on the clinical research. 

In the revised paper, age, sex, body mass index, time from hospitalization to ICU admission, time from hospitalization to mechanical ventilation initiation, APACHE II score, modified early warning score, CURB-65, transferred from skilled nursing facility, CCI, aspiration pneumonia, diabetes, congestive heart failure, myocardial infarction, chronic pulmonary disease, liver disease, moderate to severe chronic kidney disease, any malignancy, rheumatic disease, dementia, cerebrovascular disease, continuous renal replacement therapy, transfusion, vasopressors and inotropes, corticosteroids, opioids, sedatives, and PEEP are used as matching variables.

For the matching process, we used nearest neighbor 1: 1 matching method with logit regression distance.

(Table 1)

Q6) Please clarify the working definition used for acute respiratory distress syndrome in the Methods. Was it Berlin definition of acute respiratory distress syndrome? Were patients with COVID-19 diagnosis included in the study?

A6) Since 2012 Berlin definition of ARDS, recent guidelines have been published based on the Berlin definition. However, in this retrospective study, the population of this study included the patients before 2012 Berlin definition of ARDS. Therefore, we enrolled ARDS patients according to the ICD-10 codes of diagnosis.

We did not include COVID-19 patients in our study population. Although, study period was overlapped with COVID-19 pandemic, our five hospitals started isolatation beds for critically ill COVID-19 patients from November 2021. We have revised the manuscript as follows:

Materials and methods (revised manuscript, page 5 lines 16~17)

ARDS was identified with an ICD-10 code of J80. 

Materials and methods (revised manuscript, page 4 lines 11)

During the study period, severe COVID-19 patients were not hospitalized in the ICU.

---

## [Decision Letter · Decision Letter 1]

24 Oct 2022

PONE-D-21-37840R1Neuromuscular blockade in mechanically ventilated pneumonia patients with moderate to severe hypoxemia: a multicenter retrospective studyPLOS ONE

Dear Dr. Kwon,

Thank you for submitting your manuscript to PLOS ONE. After careful consideration, we feel that it has merit but does not fully meet PLOS ONE’s publication criteria as it currently stands. Therefore, we invite you to submit a revised version of the manuscript that addresses the points raised during the review process.

We look forward to receiving your revised manuscript.

Kind regards,

Steven Eric Wolf, MD

Academic Editor

PLOS ONE

Journal Requirements:

Additional Editor Comments (if provided):

Editor - Thank you for resubmitting your paper. As promised, I sent it back to the original referees who are now almost completely satisfied save a few issues. Please carefully consider the comments below and reply directly to each in a cover letter with appropriate marked and linked changes to the manuscript. I look forward to receiving the next version which I will handle personally for timeliness.

Reviewers' comments:

Reviewer's Responses to Questions

**Comments to the Author**

1. If the authors have adequately addressed your comments raised in a previous round of review and you feel that this manuscript is now acceptable for publication, you may indicate that here to bypass the “Comments to the Author” section, enter your conflict of interest statement in the “Confidential to Editor” section, and submit your "Accept" recommendation.

Reviewer #1: All comments have been addressed

Reviewer #2: All comments have been addressed

2. Is the manuscript technically sound, and do the data support the conclusions?

Reviewer #1: Yes

Reviewer #2: Yes

3. Has the statistical analysis been performed appropriately and rigorously? 

Reviewer #1: Yes

Reviewer #2: Yes

4. Have the authors made all data underlying the findings in their manuscript fully available?

Reviewer #1: Yes

Reviewer #2: No

5. Is the manuscript presented in an intelligible fashion and written in standard English?

Reviewer #1: Yes

Reviewer #2: Yes

6. Review Comments to the Author

Reviewer #1: The use of a sedation score that is not universally used in their study is a limitation, but the data and conclusions are sound and supported by what they have observed.

Reviewer #2: Thank you for addressing the comments. The authors have made appropriate adjustments to the original submission, refining and strengthening it to good effect. I have only one concern regarding excluding patients with ARDS diagnoses since it is impossible to exclude some patients with missing ARDS diagnoses remained in the included patients due to the retrospective study design. In the LUNG SAFE study, the largest international cohort among patients with ARDS, the diagnosis of ARDS was missed entirely in 40% of patients, while ARDS recognition ranged from 51% in mild ARDS to 79% in severe ARDS. In this respect, I suggest the authors keep patients with ARDS diagnosis recognition in the analysis or present both analyses, including and excluding patients with ARDS diagnosis recognition.

7. PLOS authors have the option to publish the peer review history of their article (what does this mean?). If published, this will include your full peer review and any attached files.

Reviewer #1: No

Reviewer #2: No

---

## [Author Response · Author response to Decision Letter 1]

27 Oct 2022

Response to Reviewer 1

Q) Reviewer #1: The use of a sedation score that is not universally used in their study is a limitation, but the data and conclusions are sound and supported by what they have observed.

A) Thank you for your thoughtful comment. Per your suggestion, we deleted the description in limitation section: “including the level of sedation (e.g., the Richmond Agitation and Sedation Scale, RASS) or delirium (e.g., the Confusion Assessment Method for the Intensive Care Unit, CAM-ICU).”

Response to Reviewer 2

Q)Reviewer #2: Thank you for addressing the comments. The authors have made appropriate adjustments to the original submission, refining and strengthening it to good effect. I have only one concern regarding excluding patients with ARDS diagnoses since it is impossible to exclude some patients with missing ARDS diagnoses remained in the included patients due to the retrospective study design. In the LUNG SAFE study, the largest international cohort among patients with ARDS, the diagnosis of ARDS was missed entirely in 40% of patients, while ARDS recognition ranged from 51% in mild ARDS to 79% in severe ARDS. In this respect, I suggest the authors keep patients with ARDS diagnosis recognition in the analysis or present both analyses, including and excluding patients with ARDS diagnosis recognition.

A) Thank you for your thoughtful commens. Per your suggestion, we reanalyzed the data including ARDS. After included of ARDS diagnosis, 90 d mortality rate was significiantly higher in NMB users (39.7% vs. 31.9%, p = 0.021). We described the revised results of the entire paper. We highlighted the main changes as yellow color. We thank you for your thoughtful comments again.

---

## [Editor Report · Decision Letter 2]

31 Oct 2022

Neuromuscular blockade in mechanically ventilated pneumonia patients with moderate to severe hypoxemia: a multicenter retrospective study

PONE-D-21-37840R2

Dear Dr. Kwon,

We’re pleased to inform you that your manuscript has been judged scientifically suitable for publication and will be formally accepted for publication once it meets all outstanding technical requirements.

Kind regards,

Steven Eric Wolf, MD

Academic Editor

PLOS ONE
---

## [Editor Report · Acceptance letter]

18 Nov 2022

PONE-D-21-37840R2 

Neuromuscular blockade in mechanically ventilated pneumonia patients with moderate to severe hypoxemia: a multicenter retrospective study 

Dear Dr. Kwon:

I'm pleased to inform you that your manuscript has been deemed suitable for publication in PLOS ONE. Congratulations! Your manuscript is now with our production department. 

Kind regards, 

on behalf of

Dr. Steven Eric Wolf 

Academic Editor

PLOS ONE